# Discordance of KRAS Mutational Status between Primary Tumors and Liver Metastases in Colorectal Cancer: Impact on Long-Term Survival Following Radical Resection

**DOI:** 10.3390/cancers13092148

**Published:** 2021-04-29

**Authors:** Francesco Ardito, Francesco Razionale, Lisa Salvatore, Tonia Cenci, Maria Vellone, Michele Basso, Elena Panettieri, Maria Alessandra Calegari, Giampaolo Tortora, Maurizio Martini, Felice Giuliante

**Affiliations:** 1Hepatobiliary Surgery Unit, Fondazione Policlinico Universitario A, Gemelli IRCCS, 00168 Rome, Italy; francesco.razionale01@icatt.it (F.R.); maria.vellone@unicatt.it (M.V.); elena.panettieri@unicatt.it (E.P.); felice.giuliante@unicatt.it (F.G.); 2Department of Translational Medicine and Surgery, Università Cattolica del Sacro Cuore, 00168 Rome, Italy; giampaolo.tortora@unicatt.it; 3Department of Medical Oncology, Fondazione Policlinico Universitario A, Gemelli IRCCS, 00168 Rome, Italy; lisa.salvatore@policlinicogemelli.it (L.S.); michele.basso@policlinicogemelli.it (M.B.); mariaalessandra.calegari@guest.policlinicogemelli.it (M.A.C.); 4Department of Pathology, Fondazione Policlinico Universitario A, Gemelli IRCCS, 00168 Rome, Italy; tonia.cenci@guest.policlinicogemelli.it (T.C.); Maurizio.Martini@unicatt.it (M.M.)

**Keywords:** KRAS mutation status, KRAS discordance, precision medicine, colorectal tumor, colorectal liver metastases, overall survival

## Abstract

**Simple Summary:**

KRAS mutational heterogeneity between primary colorectal cancer and liver metastases may present a challenge in assessing prognostic information prior to the multimodal treatment. Aim of our study is to assess the incidence of KRAS discordance in a single-center series by comparing primary colorectal tumor specimens with the corresponding liver metastasis. Mutation analyses in all patients were performed at the same institution. Impact of discordance on overall survival and risk factors associated with discordance were analyzed. Our study showed that KRAS discordance was observed in 15.9% of patients. In patients with wild-type colorectal primary tumors, subsequent acquisition of mutation in the corresponding liver metastasis was associated with worse overall survival and was significantly more frequently found in patients with multiple liver metastases. These results suggested that, in the era of precision medicine, the possibility of KRAS discordance should be taken into account within the multidisciplinary management of patients with metastatic colorectal cancer.

**Abstract:**

If KRAS mutation status of primary colorectal tumor is representative of corresponding colorectal liver metastases (CRLM) mutational pattern, is controversial. Several studies have reported different rates of KRAS discordance, ranging from 4 to 32%. Aim of this study is to assess the incidence of discordance and its impact on overall survival (OS) in a homogenous group of patients. KRAS mutation status was evaluated in 107 patients resected for both primary colorectal tumor and corresponding CRLM at the same institution, between 2007 and 2018. Discordance rate was 15.9%. Its incidence varied according to the time interval between the two mutation analyses (*p* = 0.025; Pearson correlation = 0.2) and it was significantly higher during the first 6 months from the time of primary tumor evaluation. On multivariable analysis, type of discordance (wild-type in primary tumor, mutation in CRLM) was the strongest predictor of poor OS (*p* < 0.001). At multivariable logistic regression analysis, the number of CRLM >3 was an independent risk factor for the risk of KRAS discordance associated with the worst prognosis (OR = 4.600; *p* = 0.047). Results of our study suggested that, in the era of precision medicine, possibility of KRAS discordance should be taken into account within multidisciplinary management of patients with metastatic colorectal cancer.

## 1. Introduction

Hepatic resection is currently the only treatment option that can offer, together with perioperative chemotherapy, a chance of long-term survival in patients with colorectal liver metastases (CRLM), resulting in 5-year survival rates of 40% [1,2], and exceeding 50% in selected patients [3,4,5]. Several clinico-pathologic factors have been used to assess prognosis following liver resection for CRLM [4,6]. However, it has been showed that these factors, often do not represent the biological heterogeneity of CRLM and are not adequate for predicting long-term outcome.

In the current era of targeted therapies and personalized medicine, molecular biomarkers have been studied as fundamental prognostic predictors that guide the type of chemotherapy and define a better selection of patients for surgery [7]. The KRAS oncogene is currently the most used molecular biomarker in patients with CRLM [8]. This mutation occurs in about 38% of colorectal tumors and in 15–35% of patients with resectable CRLM, involving mainly codons 12 and 13 in more than 95% of cases [9]. A mutation in KRAS is associated with resistance to treatment with epidermal growth factor receptor (EGFR) antibodies, showing a lower response rate to therapy. For this reason, KRAS mutation is associated with lower overall and disease-free survival following liver resection and with higher risk of relapse in the lungs [10,11].

However, it has been showed that resistance to anti-EGFR therapy may also occur among patients without KRAS mutation in colorectal tumors (KRAS wild-type). The discordance of KRAS mutation status between the primary colorectal tumor and the corresponding CRLM may represent a possible explanation for the resistance to monoclonal antibodies targeted therapies.

Generally, metastases have similar mutations to those of the corresponding colorectal primary tumor. For these reasons, in the clinical practice, the KRAS mutation status information is generally obtained from primary colorectal tumor (surgical resection or biopsy). However, if KRAS mutation status of primary tumor may be representative of the corresponding CRLM mutational pattern, is debatable. Indeed, additional mutations may occur, determining the heterogeneity between primary tumor and CRLM. It has been hypothesized that primary tumor and the corresponding CRLM may show a discrepancy in mutational pattern. This event may represent the predominant cause of resistance to therapy. In this situation, it is clear that the only evaluation of KRAS mutation status in primary colorectal tumor may be inadequate to predict response to anti-EGFR therapy of the corresponding CRLM and may provide limited information prior to multimodal treatment. Several studies have been published focusing on the incidence of discordance between primary colorectal tumor and metastases, with controversial results. Some studies reported 100% of concordance [12,13]. On the other hand, several studies have reported different rates of discordance, ranging from 4 to 32% [14,15,16,17,18,19]. This heterogeneity of results may be due to the bias of inclusion criteria in several studies, where the KRAS mutation status of primary tumor is compared with a wide variety of metastatic sites.

The aim of this study is to assess the incidence of discordance in a single-center series where all the primary colorectal tumor specimens were evaluated at the same institution and were compared with only one type of corresponding metastatic site (the liver), also evaluated at the same institution. The impact of discordance on overall survival (OS) and risk factors associated with discordance occurrence were analyzed.

## 2. Materials and Methods

### 2.1. Inclusion Criteria

This study included patients with histologically confirmed colorectal cancer who underwent primary tumor resection at our university hospital and simultaneous or delayed liver resection for corresponding CRLM at our unit between January 2007 and December 2018. Data were retrospectively extracted from a prospectively collected database established at our unit in January 1987 for all consecutive admissions related to possible liver resection. The inclusion criteria were as follows: availability of KRAS mutation analysis performed at our university hospital, of both primary colorectal tumor and the corresponding CRLM; complete resection of all CRLM; absence of unresectable extrahepatic disease; a minimum follow-up ≥2 years.

The following data were collected for each patient: demographics; site of primary tumor; primary tumor nodal involvement; size, number and distribution of CRLM; type of CRLM (synchronous or metachronous); use of preoperative chemotherapy (type, number of courses, use of targeting agents). As reported in previous studies [20,21], the location of primary tumor was classified as right-sided for tumors located between cecum and transverse colon. Left-sided tumors included those located between splenic flexure and sigmoid colon. The third group of patients included rectal tumors.

Operative details included: type of liver resection and radicality of liver resection. When the surgical free-margin was zero mm, or there was exposed tumor along the transection plane, liver resection was classified as R1-resection.

Late results included 5-year overall survival (OS).

### 2.2. Preoperative Assessment

All the patients were evaluated in our center by a multidisciplinary team including surgeons, oncologists, and radiologists. In our policy there were no predefined criteria of unresectability with regard to number, size, and bilaterality of CRLM [22]. Patients were defined resectable when all disease could be removed, leaving adequate liver remnant [23]. Technical unresectability was defined as inadequate liver remnant, or impossibility to remove all CRLM either by one- or two-stage procedure. An anticipated risk of R1 resection was not a contraindication to liver resection, although our preferred policy has always remained to obtain a tumor-free margin ≥1 cm whenever possible [24].

### 2.3. Preoperative Chemotherapy

Indications to preoperative chemotherapy were: initially unresectable CRLM or marginally resectable CRLM (high risk of R1 resection due to number, size or ill location of CRLM; ≥3 synchronous CRLM) [22]. Response to chemotherapy was assessed every 2 months (4 courses) by using the Response Evaluation Criteria in Solid Tumors [25].

### 2.4. Surgical Procedure

Liver resections were defined according to the IHPBA terminology [26]. Resections of three or more segments were classified as major hepatectomies. The surgical technique used in our unit for liver resection has been described previously [27]. Briefly, parenchymal transection was performed by the Cavitron ultrasonic surgical aspirator (CUSA 200; Valleylab, Boulder, CO, USA) and wet bipolar forceps; hemostasis and biliostasis were obtained with absorbable clips (Absolok AP200 and AP300, Johnson & Johnson Medical SpA, 00071, Roma, Italy); or with 3/0–4/0 absorbable stitches and unabsorbable ones on hepatic veins branches.

### 2.5. KRAS Mutation Analysis

KRAS mutation status analysis is routinely performed in the clinical praxis from 2007 at our University Hospital for all patients undergoing colorectal resection of primary colorectal tumor and liver resection for colorectal metastases.

All the KRAS mutation analyses were performed in both primary colorectal tumor and the corresponding CRLM at the Anatomic Pathology Unit of our University Hospital. In case of multiple CRLM, KRAS mutation analysis was performed on the largest lesion in size.

Tumor was identified in hematoxylin- and eosin-stained sections of formalin-fixed, paraffin-embedded archivial blocks. DNA was extracted from three 10 µm-slides of paraffin-embedded tissue using the QIAamp DNA mini Kit (Qiagen, Milan, Italy). In order to minimize the contamination by normal cells, the tumor areas dissected for DNA and RNA extraction contained at least 70% of tumor cells. As previously described [28,29,30], KRAS codons 12 and 13 were amplified in one PCR. Thermal cycling conditions were: 95 °C for 12 min followed by 40 cycles of 95 °C for 10 s, 55 °C for 20 s and 72 °C for 20 s. PCR conditions were as follows: primer concentration 200 nmol L^−1^, MgCl2 concentration 2 mmol L^−1^; 30 ng of genomic DNA and 12.5 µL of Eppendorf Prime mastermix (Eppendorf, Milan, Italy) in a final reaction volume of 25 µL. PCR products were electrophoresed in a 2.5% agarose gel, stained with ethidium bromide and visualized under UV light. Thereafter, 5 µL of PCR product was treated with ExoSAP-IT (GE Healthcare, Milan, Italy) following the manufacturer’s protocol, amplified with the BigDye Terminator version 3.1 cycle sequencing kit (Applied Biosystems, Milan, Italy) using the same primers of the amplification, and sequenced with an ABI PRISM 3100-Avant Genetic Analyzer (Applied Biosystems).

Each sample tested for KRAS mutational status contained at least 70% tumor cells. This percentage was assessed by the pathologist by identifying tumor cells with hematoxylin and eosin. All the analyzed samples were studied using the same technique which has a sensitivity of 15%.

KRAS discordance was defined when the KRAS mutation status (wild-type vs. mutated) in the primary colorectal tumor was different from that in the corresponding CRLM.

### 2.6. Primary Outcome

The primary outcome was the incidence of KRAS discordance and its impact on overall survival following surgical resection of both primary colorectal tumor and corresponding CRLM.

### 2.7. Secondary Outcome

The secondary outcome was the definition of risk factors associated with KRAS discordance.

### 2.8. Statistical Analysis

Continuous variables were reported as medians and interquartile ranges (IQR). Categorical variables were expressed in numbers and percentages. Chi-squared test was used for comparing categorical variables. To define the correlation between the incidence of KRAS discordance and the time interval between the KRAS mutation analysis of the primary colorectal tumor and the KRAS mutation analysis of the corresponding CRLM, the Pearson correlation coefficient was calculated with 2-tailed test of significance.

The overall survival (OS) was calculated from the date of liver resection until the date of death or censored at the last follow-up. Survival curves were generated using the Kaplan–Meier method and compared with the log-rank test. A multivariable regression analysis was performed to identify the independent prognostic factors for OS, using a Cox proportional hazards model with backward elimination for variables with *p* < 0.2 in univariate analysis. Logistic regression was used to determine the independent predictors of KRAS discordance associated with worse OS (KRAS wild-type in primary tumor and mutation in CRLM). A preliminary univariable model was created. All the variables showing a *p* < 0.2 were used for constructing the multivariable model. Odds ratio (OR), and 95% confidence intervals (95% CIs) were reported. In all the analyses, a *p* < 0.05 was considered statistically significant. Analyses were carried out with SPSS 23.0 Software (SPSS Inc., Chicago, IL, USA).

## 3. Results

Between January 2007 and December 2018, a total of 957 liver resections were performed for CRLM at our unit. Among these, 728 were primary hepatectomies (first liver resection). In such patients, the KRAS mutation analysis was available only for the CRLM in 551 patients and both for the primary colorectal tumor and the corresponding CRLM, in 107 patients, who are the subjects of our study. Characteristics observed in the study population are reported in Table 1. Primary tumor location was right-sided in 29 patients (27.1%), left-sided in 46 (43.0%), and rectum in 32 (29.9%). The median number of resected CRLM was 3 (1–20). Preoperative chemotherapy was administered in 80 patients (74.8%). The median number of courses was 9 (1–28). Administration of targeting agents was associated in 59 of the 80 patients who underwent preoperative chemotherapy (73.7%).

### 3.1. KRAS Mutation Analysis

In 94 patients (87.8%) liver resection for CRLM was performed following resection of the primary tumor, after a median time of 11 months (1–110). In 10 patients (9.4%) liver resection was performed simultaneously with primary tumor resection. In 3 patients (2.8%) liver resection was performed before primary tumor resection (liver first approach).

Overall, 107 specimens from primary colorectal tumor were analyzed together with their corresponding specimens from CRLM, to analyze the concordance in KRAS mutation status.

#### 3.1.1. Incidence of KRAS Mutation

Among the 107 patients, 33 (30.8%) had a KRAS mutation in the primary tumor and 36 (33.6%) had a KRAS mutation in the corresponding CRLM (Table 2). The most frequent mutations were found in codon 12. Type of KRAS mutations were not significantly different between primary tumors and CRLM (Table 2).

#### 3.1.2. Incidence of KRAS Discordance

Sixty-four patients (59.8%) were KRAS wild-type in both primary tumor and CRLM and 26 (24.3%) were KRAS mutated in both primary tumor and CRLM (Figure 1). Discordance was documented in 17 patients (15.9%): in 10 patients (9.4%) KRAS was wild-type in primary tumor and mutated in CRLM and in 7 patients (6.5%) KRAS was mutated in primary tumor and wild-type in CRLM (Figure 1).

#### 3.1.3. KRAS Discordance and Preoperative Chemotherapy

Twenty-seven patients (25.2%) did not undergo preoperative chemotherapy. Among these patients, the incidence of discordance was 18.5% (5 patients), not significantly different than that observed in patients who underwent preoperative chemotherapy (15.0%, 12/80 patients; *p* = 0.665).

Among the 74 patients with wild-type colorectal primary tumor, 33.9% (25 patients) underwent preoperative administration of anti-EGFR agents: 2 patients before simultaneous colorectal and liver resection and the other 23 patients prior to liver surgery. Among these 74 patients, the incidence of discordance was 13.5% (10 patients), not significantly different than that observed among the 33 patients with KRAS-mutated colorectal tumor (21.2%, 7 patients; *p* = 0.314). Forty-nine patients with wild-type colorectal primary tumor did not undergo preoperative administration of anti-EGFR agents. In such patients the incidence of discordance was 18.4% (9/49).

The incidence of KRAS discordance varied according to the time interval between the KRAS mutation analysis of the primary colorectal tumor and the mutation analysis of the corresponding CRLM (Figure 2). This incidence showed a statistical significant negative linear correlation with the time interval between the two analyses (*p* = 0.025; Pearson correlation = 0.2).

### 3.2. Overall Survival

After a median follow-up of 35.5 months (3–117), 68 patients were alive at the last follow-up. The 5-year OS for the total group of 107 patients was 55.8% (median OS: 74 months). On multivariable analysis, independent predictors of poor OS were KRAS discordance (wild-type in primary tumor, mutation in CRLM), *p* < 0.001; R1 liver resection, *p* = 0.002; primarily unresectable CRLM, *p* = 0.012 and bilobar CRLM, *p* = 0.016 (Table 3).

### 3.3. Predictors of KRAS Discordance

A multivariable logistic regression analysis for the risk of KRAS discordance associated with the worst prognosis (wild-type in primary tumor, mutation in CRLM) was performed. The number of CRLM >3 was an independent risk factor for this type of KRAS discordance (OR = 4.600; 95%CI = 1.020–20.734; *p* = 0.047). Preoperative Bevacizumab administration showed a protective effect (OR = 0.072; 95%CI = 0.007–0.716; *p* = 0.025) (Table 4).

A multivariable logistic regression analysis for the risk of KRAS discordance associated with the worst prognosis (wild-type in primary tumor, mutation in CRLM) was also performed for a sub-group of patients (49 patients) who did not undergo administration of anti-EGFR agents. In this analysis no independent risk factor for discordance were detected.

## 4. Discussion

Our study showed that discordance of KRAS mutation status between primary colorectal tumor and the corresponding CRLM does not represent a rare event. Among the 107 patients resected for both the primary tumor and CRLM at the same institution, the specimens analysis, performed by the same anatomic pathology unit, with the same technologies, documented a discordance incidence of 15.9% (17 patients).

During the last years, somatic gene mutation analyses have been increasingly used to evaluate the biology of colorectal cancer in patients undergoing resection of CRLM. To date, KRAS mutation status is considered the most well recognized prognostic biomarker to stratify prognosis among patients undergoing liver resection for CRLM [7,8]. Indeed, KRAS mutations are documented in about 15–35% of patients resected for CRLM [8]. It is well-known that KRAS mutations in patients with colorectal cancer, are associated with no response to anti-EGFR therapies [31] and with consequent significantly lower OS and disease-free survival, due to higher risk of tumor relapse, especially in the lungs. The European Society for Medical Oncology (ESMO) guidelines recommended the KRAS mutation status analysis before deciding to perform anti-EGFR treatment [32]. For these reasons, in the clinical practice, the KRAS mutation status, obtained by specimens analysis of the primary colorectal tumor (tumor biopsy or colorectal resection) represents one of the most important driver for the chemotherapy regimen choice, for the type of targeting agent to be administered in association with chemotherapy and for the subsequent surgical plan. In other words, in the majority of patients with metastatic colorectal cancer, the type of oncological and surgical management is determined by the KRAS mutation status analyzed in the primary tumor. However, on the other hand, about 50% of patients with wild-type KRAS colorectal cancer may not respond to anti-EGFR therapies. Several studies have focused on tumor diversity within the same patient, between the primary tumor and metastatic sites (intra-tumor heterogeneity) [33]. The intra-tumor heterogeneity of metastatic colorectal cancer may represent a possible explanation of resistance to therapy. Multiple subclonal driver events result in a heterogeneous tumor. This heterogeneity suggests that the analysis of primary tumor may not accurately reflect the biology of a liver metastasis. Differences in gene expression profiles may be documented in different stages of the tumor in the same patient and also within the same lesion [34].

Several studies have addressed the concordance in KRAS mutation status between primary colorectal tumors and their metastases with controversial results. While some studies have reported 100% concordance [12,13], other studies reported an incidence of discordance varying from 4 to 32% [14,15,16,17,18,19,35]. However, most of these studies collected the results of small numbers of sample analyses, often including different and heterogeneous metastatic sites and by using techniques with low sensitivity [32].

Our study analyzed the results of KRAS mutation analysis performed in a homogenous group of 107 patients, all resected at the same institution for both primary tumor and CRLM. Furthermore, all the specimens of both primary tumor and CRLM were analyzed by the same anatomic pathology unit with the same technique [28,29,30].

The overall rate of discordance in our study was 15.9%. Pathogenesis of malignant tumors involves multiple genes and sequential steps. Different tumor clones may be present in the same primary colorectal tumor and during disease progression, the mutational profile may be modified. It has been postulated that mutation may be acquired by the CRLM in a later stage, during disease progression [36]. In our study, after assessing the interval time between the two KRAS analyses (primary tumor and corresponding CRLM), it was interesting to note that intra-tumoral heterogeneity was significantly more frequently documented during the first 6 months from the time of primary tumor evaluation. Indeed, the KRAS discordance incidence showed a statistical significant negative linear correlation with the time interval between the analysis of primary tumor and the analysis of the corresponding CRLM. Discordance was significantly higher in patients who underwent simultaneous resection of both colorectal primary tumor and CRLM (30.0%) and in patients who underwent delayed resection of CRLM at an interval time not exceeding 6 months after colorectal resection (36.8%). The discordance rate was significantly lower when CRLM were resected by a delayed approach with an interval time of more than 7 months. It could mean that intra-tumoral heterogeneity was not an event occurring late in time, in patients with metachronous CRLM, but it was more frequently found in patients with synchronous CRLM managed during the first 6 months after colorectal resection. This result may be clinically relevant due to the high rate of patients with synchronous CRLM observed and treated in our series (66.3%). Indeed in such patients, most mutational analyses were performed in primary tumors and the consequent management was performed according to the primary tumor mutation status. However, in our experience, this analysis may be inadequate in about 15% of discordant patients, confirming the utility of KRAS evaluation not only in primary tumor but also in the corresponding CRLM, mostly in patients classified as KRAS wild-type but not responding to anti-EGFR therapies.

Interestingly enough the type of discordance was more frequently due to the detection of a mutation in CRLM when the primary tumor was wild-type (9.4% of patients in our series), when compared to the reverse discordance (6.5% of patients in our series). Our results arouse the issue of molecular profilation even at liver level in the view of a chemotherapeutic regimen with anti-EGFR agents, not only before liver surgery, but also in the post-operative setting. This last topic is of particular interest, since some reports suggest the possibility of a detrimental effect of anti-EGFR therapies when a mutation is detected or in the adjuvant setting even in wild-type patients [37].

Moreover, in our study, the differentiation of type of discordance was clinically relevant because at the multivariable analysis of OS, it was the strongest prognostic factor for OS. Indeed, patients with wild-type primary tumor who acquired mutation in the corresponding CRLM presented a significantly lower 5-year OS (*p* < 0.001), even worse of those carrying a KRAS mutation of both primary tumor and CRLM.

Finally, by a multivariable logistic regression analysis, we tried to identify the risk factors for KRAS discordance associated with the worst prognosis (wild-type in primary tumor, mutation in CRLM). Multiple CRLM (>3 lesions) was an independent risk factor for this type of KRAS discordance (OR = 4.600; *p* = 0.047). This was a clinically relevant result because in our series, 36.5% of patients with more than 3 CRLM, underwent liver resection. On the other hand, preoperative Bevacizumab administration showed a protective effect (OR = 0.072; *p* = 0.025).

Our results highlight that the only evaluation of KRAS mutation status in primary colorectal tumor may be not always adequate to predict response to anti-EGFR therapy of the corresponding CRLM. On the other hand, these results should be confirmed by larger series before suggesting to completely change the multidisciplinary management in the clinical praxis. To date, a routine liver biopsy of the corresponding CRLM should not be recommended. However, the mutation analysis of the corresponding CRLM may be indicated before starting chemotherapy in selected patients presenting with advanced border-line resectable disease, or in patients not responding to anti-EGFR therapy, before deciding subsequent management.

The present study has some limitations. The mutational analysis focused only on the KRAS oncogene. RAS analysis (NRAS and KRAS) together with BRAF analysis was available for all resected patients after 2013 according to the results of the PRIME study [38]. Moreover, among the 74 patients with wild-type primary tumor, 33.9% received anti-EGFR treatment in combination with chemotherapy, before liver resection. At the multivariable logistic regression analysis, administration of cetuximab was not an independent predictor of discordance. However, due to the relatively small number of cases, we cannot rule out that the observed discordance, partially at least, could be related to “de novo” KRAS mutations as a consequence of the “treatment pressure” of anti-EGFR agents [39,40,41]. In addition, even the possibility of a drug-selected (not anti-EGFR) clonal expansion of RAS mutated cells has to be considered, due to the reduced chemosensitivity of RAS-mutated clones. Actually there is no way to discriminate, in our series, between a “de novo” mutation or a clonal expansion. Nevertheless, the issue of liver metastases profilation, deserves further investigation not only in patients with a liver-limited disease but also in patients with a more advanced disease as data on “liquid biopsy guided” cetuximab re-introduction in anti-EGFR pretreated patients, suggest [42,43].

## 5. Conclusions

Our study showed that discordance of KRAS mutation status between primary colorectal tumor and the corresponding CRLM was observed in 15.9% of patients. KRAS discordance with acquisition of mutation in the corresponding CRLM was associated with significantly lower 5-year OS and was significantly more frequently found in patients with multiple CRLM.

The results of our study suggested that, in the era of precision medicine, the possibility of KRAS discordance should be taken into account within the multidisciplinary management of patients with metastatic colorectal cancer.

## Figures and Tables

**Figure 1 cancers-13-02148-f001:**
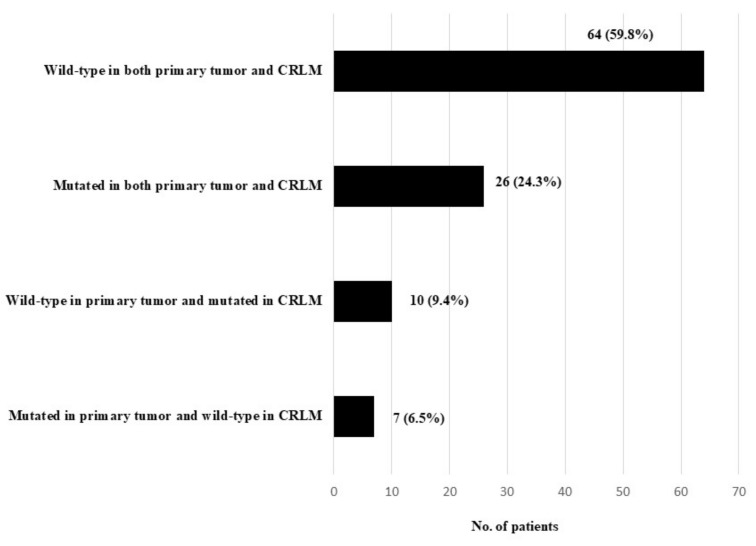
KRAS mutation status in primary colorectal tumor and corresponding CRLM.

**Figure 2 cancers-13-02148-f002:**
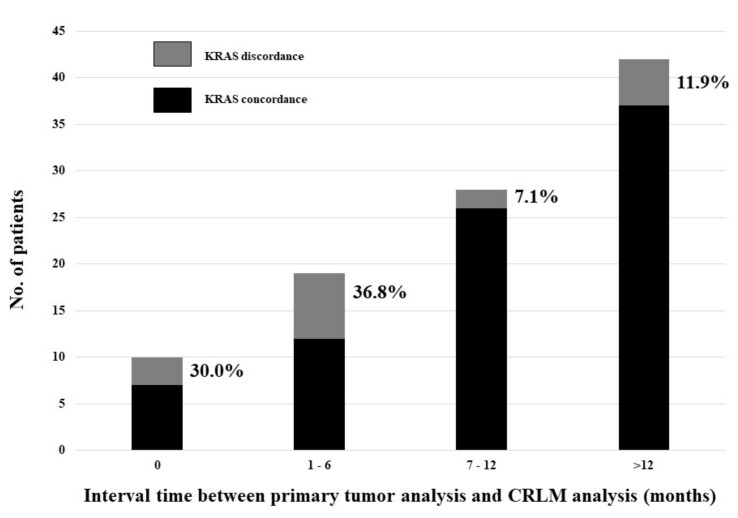
Incidence of KRAS discordance according to the time interval between the KRAS mutation analysis of the primary colorectal tumor and the mutation analysis of the corresponding CRLM.

**Table 1 cancers-13-02148-t001:** Characteristics of the 107 patients with KRAS mutation analysis of the primary tumor and the corresponding CRLM.

Variable	No. (%)
Age, median (IQR)	63 (32–83)
Gender	
Male	67 (62.6)
Female	40 (37.4)
Primary tumor	
*Location*	
Right-sided	29 (27.1)
Left-sided	46 (43.0)
Rectum	32 (29.9)
*N stage*	
N0	31 (29.0)
N1	76 (71.0)
Liver metastases	
*Timing of diagnosis*	
Synchronous	71 (66.3)
Metachronous	36 (33.7)
*Largest size* (cm)	
<5 cm	87 (81.3)
≥5 cm	20 (18.7)
*No. of metastases*	
≤3 metastases	68 (63.5)
>3 metastases	39 (36.5)
*Distribution*	
Unilobar	54 (50.5)
Bilobar	53 (49.5)
*Initial resectability*	
Resectable	93 (86.9)
Unresectable	14 (13.1)
Preoperative chemotherapy	80 (74.8)
Oxaliplatin-based	43 (53.75)
Irinotecan-based	28 (35.0)
Both	5 (6.25)
Other	4 (5.0)
*No. of cycles*	
≤6	24 (30.0)
>6	56 (70.0)
*No. of lines*	
1	74 (92.5)
>1	6 (7.5)
Associated targeting agents	59/80 (73.7)
Bevacizumab	33/59 (55.9)
Cetuximab	26/59 (44.1)
Radiological clinical response	
Partial	58 (72.5)
Stabilization	12 (15.0)
Progression	10 (12.5)
Operative features	
*Major hepatectomy*	
Yes	29 (27.1)
No	78 (72.9)
*Radicality of liver resection*	
R0	75 (70.1)
R1	32 (29.9)

Italics: sub-heading.

**Table 2 cancers-13-02148-t002:** Distribution of KRAS mutation type in primary colorectal tumor and in CRLM.

KRAS Mutation Analysis	Primary Tumor, No. (%)	CRLM, No. (%)	*p*
KRAS mutation	33 (30.8)	36 (33.6)	
Codon 12	16/33 (48.5)	21/36 (58.3%)	0.412
p.(Gly12Val)	6	7	
p.(Gly12Asp)	4	7	
p.(Gly12Ser)	2	2	
p.(Gly12Cys)	2	2	
p.(Gly12Ala)	2	2	
p.(G12R/S/C)	-	1	
Codon 13	7/33 (21.2)	7/36 (19.4)	0.855
p.(Gly13Asp)	7	7	
p.(Ala146Thr)	6/33 (18.1)	5/36 (13.9)	0.626
p.(Gln22Lys)	2/33 (6.1)	2/36 (5.6)	0.928
p.(Ala59Xaa)	-	1/36 (2.8)	
p.(Gln61Xaa)	2/33 (6.1)		

**Table 3 cancers-13-02148-t003:** Univariate and multivariable analysis of OS in 107 patients.

Variable	No. (%)	5-Year OS (%)	Univariate Analysis	Multivariable Analysis	
			*p* Value	HR (95%CI)	*p*
Age (years)					
<70	79 (73.8)	61.3	0.365		
≥70	28 (26.2)	41.7			
Gender					
Male	67 (62.6)	60.5	0.359		
Female	40 (37.4)	48.3			
Primary tumor					
*Location*					
*Right-sided*	29 (27.1)	36.1	0.048		
*Left-sided*	46 (43.0)	49.8			
*Rectum*	32 (29.9)	63.1			
*N stage*					
N0	31 (29.0)	63.0	0.338		
N1	76 (71.0)	52.2			
Liver metastases					
*Timing of diagnosis*					
Synchronous	71 (66.3)	50.3	0.073		
Metachronous	36 (33.7)	63.4			
*Largest size* (cm)					
<5	87 (81.3)	59.2	0.201		
≥5	20 (18.7)	38.0			
*No. of metastases*					
≤3	68 (63.5)	53.7	0.014		
>3	39 (36.5)	40.8			
*Distribution*					
Unilobar	54 (50.5)	69.2	0.001	2.550 (1.188–5.475)	0.016
Bilobar	53 (49.5)	38.9			
*Initial resectability*					
Resectable	93 (86.9)	60.0	<0.001	3.317 (1.297–8.483)	0.012
Unresectable	14 (13.1)	28.8			
Preoperative chemotherapy					
Yes	80 (74.8)	54.8	0.339		
No	27 (25.2)	58.0			
Irinotecan-based	28/80 (35.0)	60.1	0.816		
Oxaliplatin-based	43/80 (53.75)	51.0			
*Targeting agents*					
Yes	59/80 (73.75)	53.3	0.270		
No	21/80 (26.25)	56.9			
*No. of cycles*					
≤6	24 (30.0)	68.4	0.241		
>6	56 (70.0)	50.0			
*No. of lines*					
1	74 (92.5)	55.5	0.929		
>1	6 (7.5)	0			
*Radiological clinical response*					
Partial/stabilization	70 (87.5)	56.8	0.434		
Progression	10 (12.5)	30.0			
Operative features					
*Major hepatectomy*					
Yes	29 (27.1)	48.0	0.169		
No	78 (72.9)	58.8			
*Radicality of resection*					
R0	75 (70.1)	69.4	<0.001	2.709 (1.452–5.056)	0.002
R1	32 (29.9)	30.8			
KRAS mutation status			0.020		0.002
wild-type in primary tumor and CRLM	64 (59.8)	63.6			
mutated in primary tumor and CRLM	26 (24.3)	38.0		1.889 (0.746–4.786)	0.180
wild-type in primary tumor, mutated in CRLM	10 (9.4)	25.0		6.332 (2.398–16.715)	<0.001
mutated in primary tumor, wild-type in CRLM	7 (6.5)	60.0		1.467 (0.320–6.723)	0.622

Italics: sub-heading.

**Table 4 cancers-13-02148-t004:** Univariable and multivariable logistic regression analysis for the risk of KRAS discordance associated with the worst prognosis (wild-type in primary tumor, mutation in CRLM).

Variable	Univariable Analysis *p*	Multivariable Analysis OR (95% CI)	*p*
Age (years) ≥70	0.018		
Male sex	0.391		
Primary tumor location			
Right-sided	0.744		
Left-sided	0.780		
Rectum	0.469		
Positive lymph nodes in primary tumor	0.775		
Synchronous CRLM	0.798		
CRLM size ≥5 cm	0.998		
>3 CRLM	0.155	4.600 (1.020–20.734)	0.047
Bilobar CRLM	0.206		
Initially unresectable CRLM	0.746		
Administration of preoperative chemotherapy	0.268		
Oxaliplatin-based chemotherapy	0.655		
Irinotecan-based chemotherapy	0.167		
Association of targeting agents	0.033		
Association of Bevacizumab	0.167	0.072 (0.007–0.716)	0.025
Association of Cetuximab	0.291		
>6 cycles of chemotherapy	0.280		
Progression after chemotherapy	0.133		
Major hepatectomy	0.598		

## Data Availability

No new data were created or analyzed in this study. Data sharing is not applicable to this article.

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
