# Peer review of "Discordance of KRAS Mutational Status between Primary Tumors and Liver Metastases in Colorectal Cancer: Impact on Long-Term Survival Following Radical Resection"

_cancers, 2021, doi:10.3390/cancers13092148_

Round 1
Reviewer 1 Report
The authors presented a single center experience on KRAS mutation in CRLM compared to primary tumours. KRAS mutation status was evaluated in 107 patients resected for both primary colorectal tumor and corresponding CRLM at the same Institution. Discordance rate was 15.9% at it was higher in the first six months. The study is interesting even if the topic has been already presented in the literature. Nevertheless, there are some concerns that need to be addressed prior to consider the paper suitable for publication.
Is there any patient in the study that is naive to chemotherapy when the KRAS analysis was performed? Did you observe discordance in these patients?
How many patients are included in the follow-up? This is not indicated.
What is your suggestion for the clinical praxis? When to perform the KRAS mutation status? Do you suggest a liver biopsy before starting the therapy? Or to perform routine liver biopsy when performing large bowel resection?
Among the 74 patients with wild-type colorectal primary tumor, 33.9% (25 patients) 215 underwent preoperative administration of anti-EGFR agents. Can you include a sub-group analysis excluding these patients?
Author Response
We thank the Reviewers for their detailed analyses of our manuscript and for their constructive criticisms and advices. We have modified the manuscript according to their comments and questions.
All the new inclusions and changes performed in the text following the comments are highlighted in bold type and underlined.
REPLY TO REVIEWER #1:
Comment: The authors presented a single center experience on KRAS mutation in CRLM compared to primary tumours. KRAS mutation status was evaluated in 107 patients resected for both primary colorectal tumor and corresponding CRLM at the same Institution. Discordance rate was 15.9% at it was higher in the first six months. The study is interesting even if the topic has been already presented in the literature. Nevertheless, there are some concerns that need to be addressed prior to consider the paper suitable for publication.
Is there any patient in the study that is naive to chemotherapy when the KRAS analysis was performed? Did you observe discordance in these patients?
Response: Details about incidence of discordance in naïve patients have been included on page 7, lines 229-232.
How many patients are included in the follow-up? This is not indicated.
Response: the number of patients alive at the last follow-up has been included on page 8, line 253-254.
What is your suggestion for the clinical praxis? When to perform the KRAS mutation status? Do you suggest a liver biopsy before starting the therapy? Or to perform routine liver biopsy when performing large bowel resection?
Response: A comment about this important topic has been included on page 12, lines 362-368.
Among the 74 patients with wild-type colorectal primary tumor, 33.9% (25 patients) underwent preoperative administration of anti-EGFR agents. Can you include a sub-group analysis excluding these patients?
Response: Details about these patients were included on page 7, lines 238-240. Results of the sub-group analysis were included on page 9, lines 266-270.
Reviewer 2 Report
Studying the discordance of KRAS mutational status between primary tumors and liver metastases in colorectal cancer and relating it to long-term survival rate is an interesting topic.
IRB approval date is stated as of July 31st, 2020. Based on the manuscript, KRAS mutation status in 107 patients were analyzed between 2007 and 2018. Why the approval date is not before 2007?
Author Response
We thank the Reviewers for their detailed analyses of our manuscript and for their constructive criticisms and advices. We have modified the manuscript according to their comments and questions.
All the new inclusions and changes performed in the text following the comments are highlighted in bold type and underlined.
REPLY TO REVIEWER #2:
Comment: Studying the discordance of KRAS mutational status between primary tumors and liver metastases in colorectal cancer and relating it to long-term survival rate is an interesting topic.
IRB approval date is stated as of July 31st, 2020. Based on the manuscript, KRAS mutation status in 107 patients were analyzed between 2007 and 2018. Why the approval date is not before 2007?
Response: KRAS mutation status analysis is routinely performed in the clinical praxis from 2007 at our Institution for all patients undergoing colorectal resection of primary colorectal tumor and liver resection for colorectal metastases. The Ethics Committee of the Catholic University of the Sacred Heart of Rome approved on July 31st, 2020 our study proposal. The study proposal consisted of a retrospective analysis, starting from 2007, of pathological data in order to assess the incidence of KRAS discordance. A more detailed information has been included on page 3, lines 135-137.
Reviewer 3 Report
In this study, Francesco Ardito et al. examined KRAS mutations in primary colorectal tumors and paired matches of liver metastasized tumors. By various statistical analyses of the KRAS mutation status and clinical outcomes in CRC patients, the authors found that there are KRAS mutation discordances between primary and liver metastasized tumors in 16% of patients and further concluded that the patients with such KRAS mutation discordances show a significantly lowered 5-year overall survival. Considering the KRAS mutation status serves as a key genomic marker for the targeted therapy of metastatic CRC patients, the overall aim of this study is somewhat interesting and clinically important. However, the manuscript is poorly written, and the data presented is not clear with many errors. The authors should rewrite the manuscript with more organized data for further consideration.
Major concerns.
1) It should be clarified (or discussed with the previous literature) whether the observed KRAS mutation discordances are caused by enrichment of the mutation during the metastasis or de novo genetic events.
2) The detailed tumor content and allele frequency of KRAS mutation in the patient samples (primary and paired tumor samples) need to be documented to rule out the possibility that the observed KRAS mutation discordance and clinical outcome is just due to the underestimated allele frequency of KRAS mutation in the primary tumor.
Author Response
We thank the Reviewers for their detailed analyses of our manuscript and for their constructive criticisms and advices. We have modified the manuscript according to their comments and questions.
All the new inclusions and changes performed in the text following the comments are highlighted in bold type and underlined.
REPLY TO REVIEWER #3:
Comment: In this study, Francesco Ardito et al. examined KRAS mutations in primary colorectal tumors and paired matches of liver metastasized tumors. By various statistical analyses of the KRAS mutation status and clinical outcomes in CRC patients, the authors found that there are KRAS mutation discordances between primary and liver metastasized tumors in 16% of patients and further concluded that the patients with such KRAS mutation discordances show a significantly lowered 5-year overall survival. Considering the KRAS mutation status serves as a key genomic marker for the targeted therapy of metastatic CRC patients, the overall aim of this study is somewhat interesting and clinically important. However, the manuscript is poorly written, and the data presented is not clear with many errors. The authors should rewrite the manuscript with more organized data for further consideration.
Response: The “Results” section has been better organized and divided by subheadings. Data have been more clearly presented.
Major concerns.
1) It should be clarified (or discussed with the previous literature) whether the observed KRAS mutation discordances are caused by enrichment of the mutation during the metastasis or de novo genetic events.
Response: A comment about this interesting issue has been included on page 12, lines 376-385. References no. 42 and 43 has been included.
2) The detailed tumor content and allele frequency of KRAS mutation in the patient samples (primary and paired tumor samples) need to be documented to rule out the possibility that the observed KRAS mutation discordance and clinical outcome is just due to the underestimated allele frequency of KRAS mutation in the primary tumor.
Response: Details about tumor content and allele frequency of KRAS mutation have been included on page 4, lines 158-161.
Round 2
Reviewer 3 Report
The issues raised by the review, but not all, were clarified in the revised manuscript.